

# Proteogenomic analyses indicate bacterial methylotrophy and archaeal heterotrophy are prevalent below the grass root zone

Cristina N. Butterfield[1], Zhou Li[2], Peter F. Andeer[3], Susan Spaulding[1], Brian C. Thomas[1], Andrea Singh[1], Robert L. Hettich[2], Kenwyn B. Suttle[4], Alexander J. Probst[1], Susannah G. Tringe[5], Trent Northen[3], Chongle Pan[2] and Jillian F. Banfield[1,3]

[1] Department of Earth and Planetary Sciences, University of California, Berkeley, CA, United States
[2] Chemical Sciences Division, Oak Ridge National Laboratory, Oak Ridge, TN, Unites States
[3] Lawrence Berkeley National Laboratory, Berkeley, CA, United States
[4] Department of Ecology and Evolutionary Biology, University of California, Santa Cruz, CA, United States
[5] DOE Joint Genome Institute, Walnut Creek, CA, United States

Corresponding author
Jillian F. Banfield,
jbanfield@berkeley.edu

## ABSTRACT

Annually, half of all plant-derived carbon is added to soil where it is microbially respired to $CO_2$. However, understanding of the microbiology of this process is limited because most culture-independent methods cannot link metabolic processes to the organisms present, and this link to causative agents is necessary to predict the results of perturbations on the system. We collected soil samples at two sub-root depths (10–20 cm and 30–40 cm) before and after a rainfall-driven nutrient perturbation event in a Northern California grassland that experiences a Mediterranean climate. From ten samples, we reconstructed 198 metagenome-assembled genomes that represent all major phylotypes. We also quantified 6,835 proteins and 175 metabolites and showed that after the rain event the concentrations of many sugars and amino acids approach zero at the base of the soil profile. Unexpectedly, the genomes of novel members of the Gemmatimonadetes and Candidate Phylum Rokubacteria phyla encode pathways for methylotrophy. We infer that these abundant organisms contribute substantially to carbon turnover in the soil, given that methylotrophy proteins were among the most abundant proteins in the proteome. Previously undescribed Bathyarchaeota and Thermoplasmatales archaea are abundant in deeper soil horizons and are inferred to contribute appreciably to aromatic amino acid degradation. Many of the other bacteria appear to breakdown other components of plant biomass, as evidenced by the prevalence of various sugar and amino acid transporters and corresponding hydrolyzing machinery in the proteome. Overall, our work provides organism-resolved insight into the spatial distribution of bacteria and archaea whose activities combine to degrade plant-derived organics, limiting the transport of methanol, amino acids and sugars into underlying weathered rock. The new insights into the soil carbon cycle during an intense period of carbon turnover, including biogeochemical roles to previously little known soil microbes, were made possible via the combination of metagenomics, proteomics, and metabolomics.

# INTRODUCTION

The terrestrial carbon reservoir is primarily distributed among grasslands (∼34%), forests (∼39%), and cultivated farms (∼17%) (*White, Murray & Rohweder, 2000*). While in forests the majority of fixed carbon is stored vegetation, most fixed carbon in grasslands is stored in soil. Thus, grassland soils are one of the most important reservoirs of terrestrial carbon on the planet. This observation has motivated many studies of processes that impact the fate of carbon compounds in grassland soil systems (*Blazewicz, Schwartz & Firestone, 2014*; *Kandeler et al., 2006*; *Mau et al., 2015*; *Reinsch et al., 2014*; *Schimel & Schaeffer, 2012*; *Verastegui et al., 2014*). Over a variety of time scales, organic detritus is either respired back to $CO_2$ or degraded into smaller molecules that are transported in solution to underlying zones (*Placella, Brodie & Firestone, 2012*). In Mediterranean climate soils, this process emits an amount of $CO_2$ equal to the annual output from other ecosystems immediately following the first Fall rain, when nutrients from senesced plants are driven downward. Carbon exported from the shallow soils provides nutrients for microorganisms in lower soil horizons and the deeper subsurface.

Understanding which microorganisms are present and the pathways by which they process carbon compounds is key to understanding the form and redistribution of soil organic matter. Soil ecologists and microbiologists have employed, and continue to employ, isolation, phospholipid-derived fatty acid (PLFA) analysis, fingerprinting with denaturing gradient gel electrophoresis (DGGE) (*Muyzer, De Waal & Uitterlinden, 1993*) and terminal fragment length polymorphism (T-RFLP) (*Osborn, Moore & Timmis, 2000*) analyses, and 16S rRNA amplification and sequencing (*Banning et al., 2011*; *He, Xu & Hughes, 2006*; *Henckel, Friedrich & Conrad, 1999*) to identify soil microbes. Bacteria reported in soils are typically affiliated with the Proteobacteria, Firmicutes, Acidobacteria, Actinobacteria, Verrucomicrobia, Gemmatimonadetes, and Bacteroidetes phyla (*Evans & Wallenstein, 2012*; *Fierer et al., 2012*; *Kuramae et al., 2012*). Given the evolutionary/adaptation pressures in different soil types (*McKissock, Gilkes & Walker, 2002*), nutrient availability (*Adair, Wratten & Lear, 2013*; *Goldfarb et al., 2011*; *Veresoglou et al., 2012*), temperature and moisture (*Aanderud et al., 2013*; *Peltoniemi et al., 2015*), pH (*Lauber et al., 2009*), variety of vegetation (*Herzberger, Duncan & Jackson, 2014*; *Piper et al., 2015*; *Prober et al., 2015*), and microenvironments within the soil, it is expected that there will be substantial genetic diversity within many of these reported phyla, giving rise to clusters of closely related organisms, as well as organisms from additional phyla that have thus-far eluded detection in soils.

Prior studies of microbial functions (i.e., ammonia oxidation, methylotrophy) in soil have used targeted approaches such as gene amplification (qPCR, pyrosequencing) (*Hofmann et al., 2016*; *Pester et al., 2012*; *Stacheter et al., 2013*), culturing of isolates and enrichments (*Beck et al., 2014*). More recently, metagenomic methods have been applied to soil samples with the objective of providing a cultivation- and primer-independent

view of microbial community composition and functional capacities (*Delmont et al., 2015*; *Hultman et al., 2015*; *Luo et al., 2014*; *White et al., 2016*). Genomes provide metabolic insight (including for organisms that have not been cultivated) and enable identification of pathways involved in biogeochemical processes, but they have rarely been reconstructed from soil (*Delmont et al., 2015*; *Hultman et al., 2015*; *Pell et al., 2012*). These recent studies have shown that resolution of the community is possible and that much of the community's metabolic potential centers around respiration, complex carbohydrate degradation and central metabolism.

Here we conducted a multi-omic investigation of the microbiology and microbial activity in the shallow sub-root and deeper regions of grassland soil that experiences a Mediterranean climate. Samples were collected during the major annual period of carbon turnover around the time of the first Fall rain event. The well-studied meadow (*Cruz-Martínez et al., 2009*; *Suttle, Thomsen & Power, 2007*) is located in the Angelo Coastal Reserve in Northern California (part of the Eel River Critical Zone Observatory). These grassland soils provide an ideal system for studying processes in the sub-root zone because roots are confined to a well-defined horizon. Prior work in this meadow documented that carbon is fixed by ∼50 plant species (*Suttle, Thomsen & Power, 2007*) during the wet spring season. The carbon compounds accumulate in the upper soil horizon after plants die late in summer (*Aerts, Bakker & De Caluwe, 1992*; *Berendse, 1994*; *Wedin & Tilman, 1990*). Thus, portions of these carbon compounds that accumulate in the upper 10 cm of the soil include both leaf litter and dead root material will percolate down as dissolved metabolites through the 40–50 cm deep soils, which are developed on weathered vermiculite-dominated argillite and sandstone.

Our sampling scheme was designed to probe microbial diversity and active carbon turnover in soil using a combined metagenomic, proteomic and metabolomic approach. An important motivation for recovery of genomes from the metagenomes is that protein sequences can be predicted in organism context and used in mass spectrometry studies to identify proteins that are highly abundant in microbial cells (*Brooks et al., 2015*; *Mosier et al., 2015*). This information, in combination with metabolite concentrations measured through the soil profile, enables identification of the organisms, pathways and spatial distribution of carbon turnover processes at the time of sample collection. We uncovered roles for bacteria and archaea from phylum lineages lacking isolated representatives and identify methylotrophy and archaeal heterotrophy as major carbon cycling processes in the sub-root zone. The study demonstrates that genome-resolved multi-omic approaches can be effectively used to interrogate microbiallymediated processes in one of the Earth's complex ecosystems.

## MATERIALS & METHODS

### Sampling and DNA extraction

We collected samples from the Angelo Coast Range Reserve (with permission under APP # 27790) meadow 39°44′21.4″N 123°37′51.0″W) on four days: before the rain, four and six days after one inch of rain fell, and two days after three inches of rain fell in September, 2013.

At two plots 10 m apart in the Northern end of the meadow, a soil pit for each sampling day was dug to 50 cm and depth was marked every 10 cm. Approximately 1 kg of soil was removed from each of two depths (10–20 cm and 30–40 cm) using sterilized stainless steel hand trowels. Each sample was homogenized briefly in a sterile bowl and divided into several sterile Whirl-Pak bags. One sample bag was placed on wet ice for transport to the lab for pH and moisture analyses. The remaining samples were immediately flash frozen in a mixture of dry ice and ethanol and then placed on dry ice for transport to the lab for long-term storage at $-80\,°C$. For pH analysis, 2 g of fresh, field-wet soil were suspended in 10 mL 0.01 M $CaCl_2$ and shaken for one hour at 100 rpm. The suspension was then centrifuged for 5 min at 6,000 rpm at $4\,°C$, and the resulting supernatant was filtered through a #1 Whatman filter and analyzed with a pH probe. Gravimetric soil moisture was determined by weighing subsamples of sieved soil (2 mm sieve) before and after drying at $60\,°C$ for at least 48 h. Soil particle size distribution was determined by measuring the relative density of the soil suspended in 5% sodium hexametaphosphate solution with a hydrometer over the course of settling (40 s–2 h). To measure extractable organic carbon content, 5 g soil samples were suspended in 25 mL of 0.5 M potassium sulfate and shaken for two hours at 150 rpm. The suspension was then filtered through a #1 Whatman filter and quantified on an OI Analytical 1010, Total Organic Carbon Analyzer. Soil characteristics are summarized in Fig. S1.

For each depth, DNA was extracted using MoBio Laboratories PowerMax Soil DNA Isolation kits from 10 g of soil from ten of a the much larger set of samples used for other analyses: (1) pre-rain in plot 1 from 10–20 cm, (2) pre-rain in plot 1 from 30–40 cm, (3) four days after the first rain in plot 1 from 10–20 cm, (4) four days after the first rain in plot 1 from 30–40 cm, (5) six days after the first rain in plot 1 from 10–20 cm, (6) six days after the first rain in plot 1 from 30–40 cm, (7) six days after the first rain in plot 2 from 10–20 cm, (8) two days after the second rain in plot 1 from 10–20 cm, (9) two days after the second rain in plot 1 from 30–40 cm, and (10) two days after the second rain in plot 2 from 10–20 cm. We optimized the protocol for our samples, to maximize DNA yield while minimizing shearing: each sample was only vortexed for 1 min, followed by a 30 min heat step at $65\,°C$, inverting every 10 min. Each sample was also extracted twice, and combined at the spin filter step. We performed two elution steps of 5 mL each, and precipitated the DNA using sodium acetate and glycogen, resuspending in 100 uL of 10 mM Tris buffer. This resulted in unsheared large fragment size DNA, with average yields of 2,351 ng/g soil (10–20 cm depth) and 1,277 ng/g soil (30–40 cm depth). Fragment size was checked on 0.5% agarose gels using a 23 kb genomic DNA ladder and DNA concentration was measured using a Qubit Fluorometric Quantitation device, dsDNA Broad Range Assay Kit.

## DNA sequencing and reconstruction of genomes

DNA sequencing was conducted at the Joint Genome Institute, USA. 250 bp paired Illumina reads were processed with BBMap (https://sourceforge.net/projects/bbmap/). BBMap was run twice (1) to trim adapters bbduk.sh was used with parameters $k = 23$, mink $= 11$, hdist $= 1$, tbo, tpe, ktrim $= r$, ftm $= 5$ and (2) to remove phiX and Illumina trace contaminants

bbduk.sh was used with parameters $k = 31$, hdist $= 1$. Illumina adapter reference sequences, the phiX genome and Illumina traces were provided by JGI.

Reads were further trimmed with Sickle (https://github.com/najoshi/sickle) using default settings. Paired end read datasets from each sample were assembled independently from one another using idba_ud under default settings, including the –pre_correction option (*Peng et al., 2012*). For scaffolds greater than 1,000 bp, open reading frames were predicted with Prodigal (*Hyatt et al., 2010*) and functional annotations were determined through similarity searches against the UniProt, UniRef90 (*Suzek et al., 2007*) and KEGG (*Kanehisa et al., 2012*; *Ogata et al., 1999*) databases. tRNAs were predicted for each scaffold using tRNAscan-SE (http://lowelab.ucsc.edu/tRNAscan-SE/). To identify 16S rRNA gene sequences, we searched all assembled scaffolds against the manually curated structural alignment of the 16S rRNA provided with SSU-Align (*Nawrocki, Kolbe & Eddy, 2009*). Coverage values for each scaffold were calculated by read mapping using Bowtie2 (http://bowtie-bio.sourceforge.net/bowtie2/index.shtml) using default settings. The scaffolds, all associated annotations, and coverage information were then processed and uploaded into ggKbase: http://ggkbase.berkeley.edu/angelo_ncbi_2016/organisms. The sequencing reads have been deposited as "Meadow soil samples from Angelo, CA genome sequencing and assembly": SRA302421—2015-10-05T12:25:59.383. Genomes are currently being processed for submission to NCBI under accession PRJNA297196.

We established a phylogenetic profile for each scaffold by comparing the genes to a database of reference genomes. Assignment of scaffolds >8 kb to genome bins was accomplished using emergent self-organizing maps (ESOM) (Fig. S5). The matrix used in the ESOM was built from a combination of series coverage patterns across samples for each scaffold (ten columns) and tetranucleotide frequency of each scaffold (256 columns). Bins were fine tuned to remove scaffolds classified as wrongly binned based on phylogenetic information or other anomalies using the visualization tools provided by ggKbase. Genome bins were named based on placement in phylogenetically informative gene trees and the overall taxonomic profile of each bin. Bin completeness was evaluated based on the recovery of content of a set of 51 single copy genes for bacteria and 38 single copy genes for archaea using a tool developed in *Probst et al. (2016)*. The phylogenetic signal, in combination with aberrant coverage and/or GC content, was used to identify bin contaminants. Draft quality genomes, defined as genome bins from metagenomes, contain at least 70% of the requisite single copy genes within minimal duplication (a firm cutoff for duplicate genes was not used because some arise due to genes split by scaffolding gaps or contig ends).

## Time series analysis

The relative coverage for every scaffold encoding a ribosomal protein S3 gene, thus representing a single strain was determined to indicate the relative abundance of each organism in each sample. Coverage values were normalized to account for differences in sample data size. Values for each species in the same phylum were summed to generate the stacked bar chart presented in Fig. 1 in the 'Results' section.

## Proteomics methods

For each soil sample, total proteins were extracted from 10 g of soil using NoviPure®
Soil Protein Extraction Kit (MoBio). The crude protein extracts were concentrated to ~1
ml using Amicon® Ultra-4 Centrifugal Filter Units (30 KDa molecular weight cut-off,
Millipore). Trichloroacetic acid was then added to precipitate proteins overnight at 4 °C.
Proteins were pelleted by centrifugation at 4 °C, washed with ice-cold acetone three times,
and re-solubilized in guanidine (6 M) and dithiothreitol (10 mM). Bicinchoninic acid
assays were conducted to estimate the protein concentration before adding dithiothreitol.
50 μg of proteins from each soil sample was further processed with the filter-aided sample
preparation (*Wisniewski et al., 2009*). Proteins were first trypsin digested overnight in an
enzyme:substrate ratio of 1:100 (weight:weight) at room temperature with gentle shaking,
followed by a secondary digestion for 4 h. All digested peptide samples were stored at
−80 °C.

LC-MS/MS proteomic measurements were carried out with 11-step online
multidimensional protein identification technology (MudPIT) (*Washburn, Wolters &
Yates, 2001*) on an LTQ Orbitrap Elite mass spectrometer (Thermo Scientific), as described
previously (*Li et al., 2014*). In each MudPIT run, 25 μg of peptides were loaded offline into
a 150-μm-I.D. two-dimensional back column (Polymicro Technologies) packed with 3 cm
of C18 reverse phase (RP) resin (Luna, Phenomenex) and 3 cm of strong cation exchange
(SCX) resin (Luna, Phenomenex). The back column loaded with peptides was de-salted
offline with 100% Solvent A (95% $H_2O$, 5% $CH_3CN$ and 0.1% formic acid) and washed
with a 1-h gradient from 100% Solvent A to 100% Solvent B (30% $H_2O$, 70% $CH_3CN$ and
0.1% formic acid) to move peptides from RP resin to SCX resin. Then, the back column
was connected to a 100-μm-I.D. front column (New Objective) packed in-house with
15cm of C18 RP resin and placed in-line with a U3000 quaternary HPLC pump (Dionex).
Each MudPIT run was configured with 11 SCX fractionations using 5%, 7%, 10%, 12%,
15%, 17%, 20%, 25%, 35%, 50% and 100% of Solvent D (500mM ammonium acetate
dissolved in Solvent A). Each SCX fraction was separated by a 110-min RP gradient from
100% Solvent A to 50% Solvent B. The LC eluent was directly nanosprayed (Proxeon)
into an LTQ Orbitrap Elite mass spectrometer (Thermo Scientific). Both MS scans and
HCD MS/MS scans were acquired in Orbitrap with the resolution of 30,000 and 15,000,
respectively. The top 10 most abundant precursor ions were selected for MS/MS analysis
by HCD after each MS scan. Peptides of each soil sample were measured in technical
duplicates.

A protein sequence database was constructed from 3,408,250 full-length predicted
proteins (combined file size of 863.56 Gb) by metagenomics from four samples; (1)
pre-rain plot 1 10–20 cm, 2 days after second rain (2) plot 1 10–20 cm, (3) plot 1 30–40
cm, and (4) plot 2 10–20 cm, and their reverse sequences as decoys for estimation of
false discovery rate (FDR) (*Elias & Gygi, 2007*). Database searching was performed with
Sipros 3.0 (*Hyatt & Pan, 2012*; *Wang et al., 2013*) on the Titan supercomputer at Oak Ridge
Leadership Computing Facility. The following parameters were used: dynamic oxidation
of methionine, static alkylation of cysteine by iodoacetamide, 0.03 Da mass tolerance
for precursor ions and 0.01 Da for fragment ions, up to three missed cleavages, and full

enzyme specificity required. The FDR was strictly controlled at the peptide level (1%). One unique peptide was required for each identified protein/protein group. Indistinguishable proteins were combined into protein groups based on the parsimony rule (*Nesvizhskii & Aebersold, 2005*). The numbers of identified protein/protein group per sample ranges is provided in Table S3B. Proteins were linked to draft genomes so the functions could be assigned to individual organisms. Spectral counts of proteins were normalized across the samples for label-free quantification as described previously (*Pan & Banfield, 2014*; *Wang et al., 2013*). The mass spectrometry proteomics data have been deposited to the ProteomeXchange Consortium via the PRIDE (*Vizcaíno et al., 2016*) partner repository with the dataset identifier PXD004965.

## Metabolomics methods

Metabolites were extracted from sieved (2 mm) soil samples using an aqueous extraction protocol (modified from *Swenson et al., 2015*). Briefly, sieved soils (4 g) were incubated (200 rpm, 1 h, 4 °C) in triplicate with MilliQ water (16 ml) amended with 1.6 µg/ml of ABMBA (2-Amino-3-Bromo-5-methylbenzoic acid) and 2 µg/ml of UL-13C-Glucose included as internal extraction standards. Samples were centrifuged (3,220 × g, 15 min, 4 °C) and the supernatant was carefully decanted. Soils were then back extracted twice with MilliQ water (4 ml, 2 ml) as above but with abbreviated incubation times (15 min and 30 s). Following centrifugation, supernatants were pooled, frozen (−80 °C) and lyophilized. Metabolites were then re-suspended in 100% methanol (250 ml, to limit salt solubility) with internal standards (5 µg/ml—Table S5) and filtered (Millipore, 0.22 mm PVDF microcentrifuge filters). Samples were analyzed in random order on an Agilent 6550 iFunnel Q-TOF LC/MS system with a SeQuant ZIC-pHILIC zwitterionic exchange column (150 × 2.1 mm, 5 mm, Merck Millipore) using a neutral pH (5 mM NH$_4$OAc)/acetonitrile gradient (10% to 55% aqueous phase over 25 min at 0.25 ml/min) in both positive and negative ionization mode. Metabolites and putative metabolites were identified manually through several methods including: comparison to the Northen Lab standards library (>290 compounds), MS/MS analyses, and formula generation (Agilent MassHunter) (Table S5). Peak detection and areas were determined using the Northen Lab's Metabolite Atlas software (http://metatlas.nersc.gov) (Table S6). Internal standards (including extraction standards) were used for quality control analyses and to detect and control for retention time shifts and other analytical variability in any of the sample analyses. A one-way ANOVA was conducted on metabolites across the samples to determine if they changed significantly between samples; *p*-values were then corrected for multiple comparisons (*Benjamini & Hochberg, 1995*) (Table S7). Significant changes were determined for metabolites between sampling times at each depth and between depths in each sample using ANOVA with post-hoc Tukey HSD pairwise analyses (Table S8).

## 16S rRNA and rpS3 phylogenetic analyses

For the 16S ribosomal RNA (rRNA) tree and ribosomal protein S3 (rpS3), alignments were generated from all 16S rRNA and rpS3 genes available the metagenomes. Sequences are provided in the Supplemental Information. All 16S rRNA genes longer than 660

bp were aligned using the SINA alignment algorithm through the SILVA web interface (*Pruesse, Peplies & Glöckner, 2012*; *Pruesse et al., 2007*). All rpS3 amino acid sequences longer than 180 aa long were aligned using MUSCLE (*Edgar, 2004a*; *Edgar, 2004b*). The full alignments were stripped of columns containing 95% or more gaps. A maximum-likelihood phylogeny was inferred using RAxML (*Stamatakis, 2014*) run using the GTRCAT model of evolution for the 16S rRNA and PROTGAMMLG for the rpS3. The RAxML inference included calculation of 300 bootstrap iterations for the 16S rRNA tree and 100 for the rpS3 tree (MRE-based Bootstopping criterion), with 100 randomly sampled to determine support values.

## Ordination analyses of microbial community structure

Ribosomal protein S3 genes were retrieved using HMMs build from the dataset published in *Hug et al. (2015)* and used to search against predicted protein sequences in all samples. Only sequences that spanned at least 60% of the alignment were included and clustered at 99% similarity (equivalent to species level *Sharon et al., 2015*) using Usearch (clusterfast, *Edgar (2010)*). Abundances of each cluster in each sample were determined from scaffold coverage (see above) and normalized to percent abundances in each sample. Principle coordinate analysis (PCoA) based on Bray-Curtis distance measure was computed using the R programming environment (*R Core Team, 2015*) in conjunction with the vegan package (*Dixon, 2003*).

## Methanol dehydrogenase tree

The amino acid sequences of the PQQ-dependent methanol dehydrogenase proteins detected in the proteomics data were aligned to reference sequences with MUSCLE and this alignment was used to build a tree with RAxML with the PROTGAMMAWAG model and 100 bootstrap iterations.

## Physical and chemical characterization of the soil

Samples of soil and the weathered bedrock (mudstone and sandstone) were collected for mineralogical and other analyses using electron microprobe, X-ray diffraction and scanning electron microscopic methods. Minerals identified included vermiculite (the predominant mineral in the soil), plagioclase and alkali feldspars, minor apatite and a mixture of Fe, Mn-oxyhydroxides.

## RESULTS

Between 14 and 25 Gb of DNA sequence data were obtained per soil sample for the ten soil samples, two of which were time and depth replicates. The 250 bp reads were assembled as detailed in 'Methods,' resulting in 2,982,775 contigs > 1,000 bp in length (42,770 contigs > 10,000 bp, the longest was 538,000 bp). In total, these contigs encode 8,773,880 genes (Table S1). For each sample, the reads were then mapped back to the assembled contigs of the sample to generate coverage statistics.

We compared the microbial community structure across the sample series using coverage of scaffolds assembled from the sequence datasets. Out of the 1,420 microorganisms

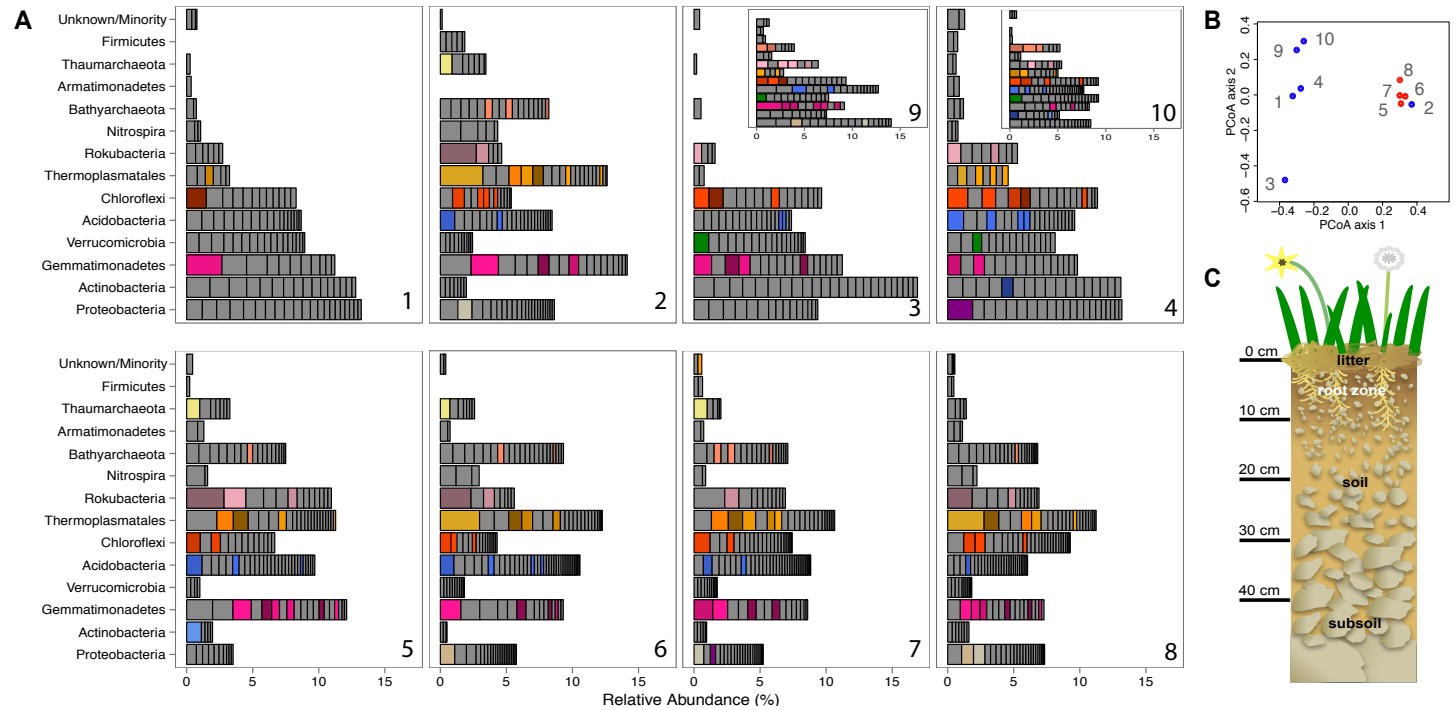

**Figure 1** **Prokaryotic diversity and abundance over sampling time and depth.** Ribosomal protein S3-encoding scaffold relative abundances for each sample are plotted in (A), and organized as follows: pre-rain (1, 5), four days after the first rainfall (2, 6), six days after the first rainfall (3, 7, 9) and two days after the second rainfall (4, 8, 10) at 10–20 cm (1–4, 9, 10) and 30–40 cm deep (5–8). Persistent and abundant species (>1% abundant in multiple samples) are colored by phylum and shaded by species. The beta diversity PCoA plot (B) compares the ten sample communities (blue denotes 10–20 cm and red denotes 30–40 cm samples). A diagram depicting the sampled ecosystem (C) shows the shallow roots of the annual forbs and grasses mostly remain in the top 10 cm and the rocks increase in size in the deeper soil.

detected based on marker gene (rpS3) sequences, 652 occurred in 2–18 times (average: 4.2 ± 2.86) over the ten samples and had relative abundances of between 0.06 and 3.2% of the community (Fig. 1). The same information was used in an ordination analysis and showed that samples from the same depth were more similar to each other than to those from the other depth with the exception of the first post-rain 10–20 cm sample (Fig. 1). The results indicate substantial overlap in the organisms present, especially among samples from the same soil depth. The same organisms were also observed in soil samples collected at the same depth and time from sites separated by ∼10 m. In fact, many of these organisms are highly abundant, ranking in the top ten most abundant organisms and amounting for > 1% relative abundance in each sample (Fig. 1). The availability of sequence information for multiple independently assembled samples with substantial overlap in community membership enabled the addition of series abundance parameterization to nucleotide frequency-based genome binning. We generated 198 bins, 46 of which were classified as metagenome-assembled draft genomes. Most sequence fragments that could not be assigned to specific genomes were grouped at the phylum-level and assigned to "megabins" (Table S2A). However, most of the relatively high coverage scaffolds in each sample were binned into draft genomes.

A notable feature of the soil microbial community compositions, evident based on phylogenetic analysis of single copy genes, is the high representation of organisms in the sub-root zone soils from phyla that are relatively poorly represented in the NCBI database. For example, we obtained draft genomes for two Gemmatimonadetes, two Verrucomicrobia, eight Acidobacteria, one Armatimonadetes, three Chloroflexi, and three Nitrospirae. Importantly, we also reconstructed seven draft genomes from the Rokubacteria, a bacterial Candidate Phylum first reported from aquifer sediment in 2015 (*Hug et al., 2016*). In addition, we reconstructed draft genomes for one Betaproteobacterium, two Deltaproteobacteria an Actinomycetales and one novel Actinobacteria. The reconstructed genomes, which represent all major phyla detected in the soil samples, substantially expand the coverage of phylogenetic diversity in the NCBI database (Table S2B). In addition, we generated four draft genomes from the ''Miscellaneous Crenarchaeota Group'' (MCG), two from the ''Soil Crenarchaeota Group'' (SCG) and three from the ''South African Gold Mine Miscellaneous Group'' (SAGMCG). The SCG and SAGMCG are likely within the Thaumarchaeotes whereas the MCG are novel Bathyarchaeota (Figs. S2A and S2B). We also sampled Euryarchaeotes from the Thermoplasmatales lineage, but the genome bins were not well resolved.

Phylogenetic trees constructed using marker genes for both bacteria and the archaea from samples collected from different depths and times display structures similar to the seed puff of a dandelion, with many closely related strains at the termini of most branches (Fig. 2 and Fig. S2). In prior studies, strains were grouped at higher taxonomic levels, up to the phylum level (*Delmont et al., 2015*; *Hultman et al., 2015*). The Gemmatimonadetes phylum branch of the ribosomal protein S3 phylogenetic tree (Fig. 2) exhibits fine scale diversity that is comparable to the observed level of diversity detected in every phylum. Many organisms of the same type (separated by zero branch length in Fig. 2) occurred in samples collected at different times and from both depths.

Before the rain, the most abundant organisms in the shallow depth interval (10–20 cm) are Gemmatimonadetes and Actinobacteria species. Gemmatimonadetes species are also abundant after the rain event. Thermoplasmatales are highly abundant in samples collected after the rain event (Fig. 1).

In general, the microbial community composition of 30–40 cm depth samples differed from that of the 10–20 cm depth samples and samples collected before and after the rain event were less different than those collected from the shallower soil. Notably, the community sampled from deeper soil prior to the rain event was dominated by a Rokubacterium and several Thermoplasmatales (Euryarchaeota) were abundant. In fact, around 20% of the microorganisms sampled in the deeper soil interval were archaea, an interesting finding because archaea are generally believed to be relatively rare compared to bacteria in soil (*Fierer et al., 2012*). In samples collected after the rain, three Thermoplasmatales archaea are highly abundant and rank in the top 10 most abundant organisms in the deeper soil (Fig. 1).

We obtained proteomic information to provide insight into the active pathways that mediated carbon and nitrogen compound transformations. Between 2,881 and 4,716 proteins/protein groups were identified per soil sample (Table S3A). Overall, we identified

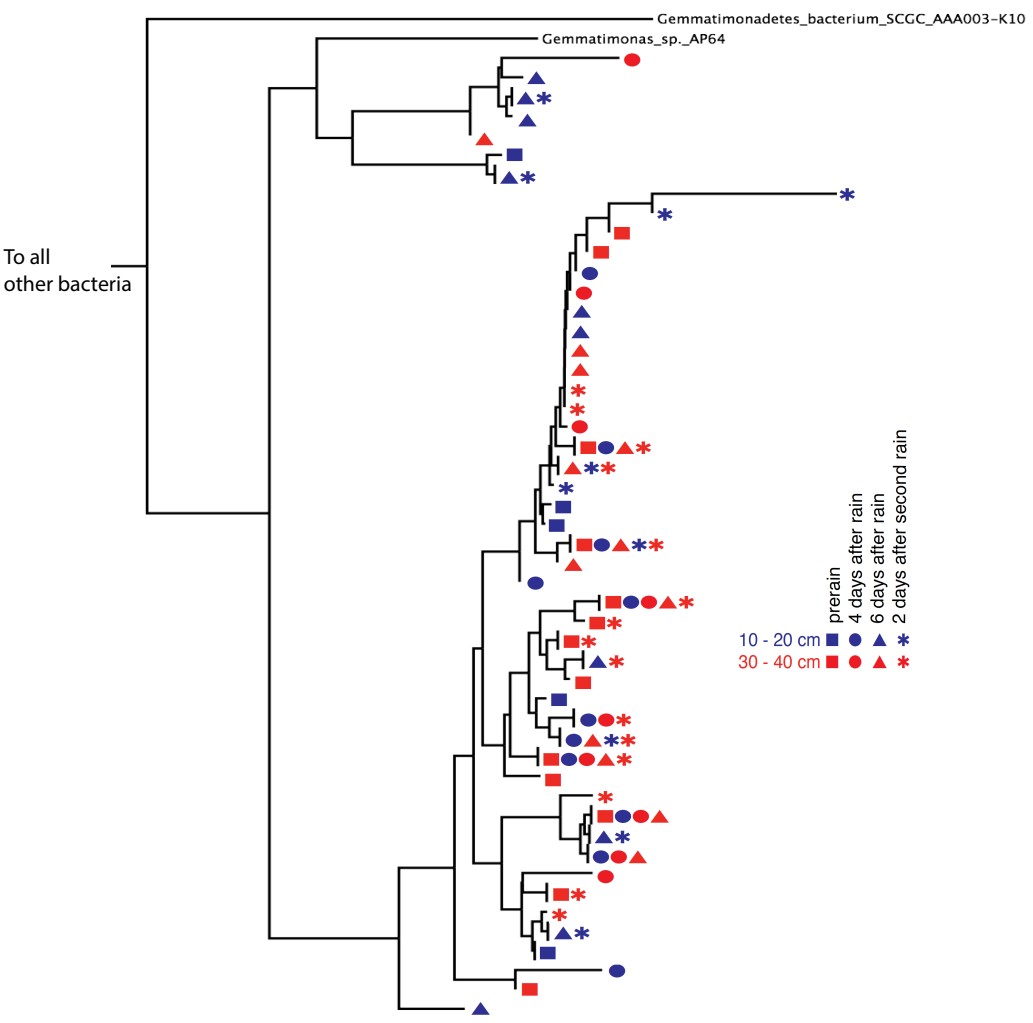

**Figure 2 Gemmatimonadetes phylogenetic ribosomal protein S3 tree.** Subsection of the experiment's ribosomal protein S3 phylogenetic tree shows typical diversity within the phyla: dozens of novel, closely related organisms of the Gemmatimonadetes bacterial phylum inhabit the two soil zones (10–20 cm, blue and 30–40 cm, red) before (squares) and after the rain events (four days after: circles, six days after: triangles, and two days after the second rain: asterisks). Identical ribosomal protein S3 sequence branches representing members of the same species were collapsed and the time and depth symbols of these members are presented horizontally.

6,835 proteins and 6,378 protein groups in the soil microbiome, based on 28,782 distinct peptide identifications. It is important to note that we could link most identified proteins to the specific microorganisms from which they derived because a significant fraction of our sequence data was genome-resolved.

Significantly, the most abundant protein in the proteome was not involved in plant sugar breakdown. Rather, it was pyrrolo-quinoline quinone (PQQ)-dependent methanol dehydrogenase (MDH). This protein is encoded in the genome of a Gemmatimonadetes bacterium. PQQ dependent MDH (PQQ MDH) is the second step of methanotrophy, and follows the oxidation of methane to methanol by methane monooxygenase. However, methane monooxygenase was not present in any of the genomes, including those harboring
the methanol dehydrogenase. Thus, we infer that this enzyme is involved in methylotrophy, the consumption of methanol. Lesser abundant PQQ MDH proteins from Rokubacteria were also detected in the proteomics analysis

MDH proteins have typically well-conserved sequences. The MDH proteins represented in the proteome were aligned with sequences from the literature (*Taubert et al., 2015*) and a tree was built, resulting in the clustering of the proteins by phylum. The Rokubacteria XoxF sequences formed a distinct branch in the MDH protein tree (XoxF and MxaF families) and the Gemmatimonadetes sequences formed a new clade with the XoxF sequences from various Proteobacteria (*Taubert et al., 2015*) (Fig. 3) The Gemmatimonadetes PQQ-MDH proteins were generally more abundant than those from Rokubacteria (as shown in the heat map on the right side of Fig. 3). The MDH proteins all contain the catalytic and cofactor binding residues required for activity (*Anthony & Williams, 2003*), including those for PQQ, as well as the aspartate residues thought to select for the lanthanides such as $Ce^{3+}$ and $La^{3+}$ (*Keltjens et al., 2014*; *Pol et al., 2014*). The selection for lanthanides over $Ca^{2+}$ is an interesting bioinorganic trait because while lanthanides are abundant in the Earth's crust, they are highly insoluble and thus considered biologically unavailable and in turn not well studied (*Skovran & Gomez, 2015*). Despite this, $La^{3+}$ is required for the activity of the XoxF type MDH perhaps because it a more efficient Lewis acid in the polarization of PQQ than $Ca^{2+}$ (*Bogart, Lewis & Schelter, 2015*; *Pol et al., 2014*).

Following oxidation of methanol by PQQ MDH the toxic formaldehyde product must be moved from the periplasm to the cytosol for transformation and incorporation into 3C compounds. The reaction can occur via one of three pathways: the glutathione, tetrahydromethanopterin (THMPT), or tetrahydrofolate (THF)-linked formaldehyde oxidation pathways (*Chistoserdova, 2011*). The Gemmatimonadetes genomes have the entire THMPT formaldehyde oxidation pathway and the tetrahydrofolate to serine pathway for these reactions. Furthermore, they encode the PQQ biosynthesis machinery. The THMPT biosynthesis machinery is encoded immediately upstream of the PQQ MDH gene (Fig. S3) (*Scott & Rasche, 2002*). The results strongly support the capacity for methylotrophy in these soil-associated Gemmatimonadetes and functioning of this pathway (Fig. 4 and Fig. S4).

Although the Rokubacteria have PQQ MDH, the genomes do not encode any of the three pathways for formaldehyde transformation. It has been suggested that the XoxF type of MDH is able to convert methanol directly to formate, bypassing the formaldehyde oxidation mechanism (*Pol et al., 2014*). Formate then may be broken down using formate dehydrogenase to yield energy or be assimilated by one of several pathways. The Rokubacteria also may be carrying out beta-oxidation of fatty acids, as these proteins for this pathway are abundant in the proteome.

We identified many proteins likely responsible for decomposing matter from the meadow's early summer senescing annual grasses (*Bromus hordeaceus, Bromus diandrus, and Bromus tectorum*) and annual lupine (*Lupinus bicolor*) (for a full species list, see Table S4). Highly represented were carbohydrate-active enzymes such as glycoside hydrolases, polysaccharide lyases, and many sugar and amino acid transport proteins (Table S3A). Notably, enzymes of the Thermoplasmatales and Bathyarchaeota archaea

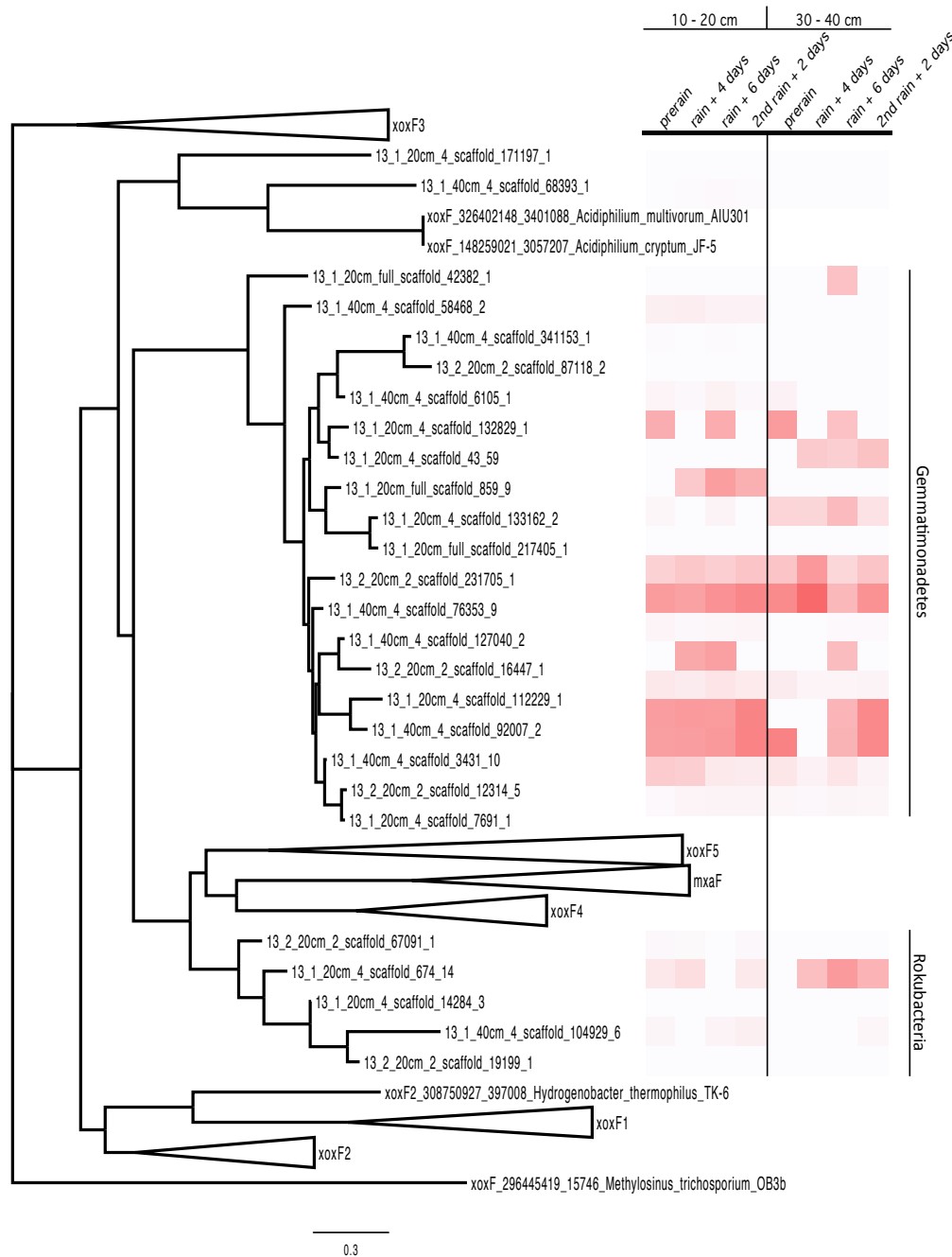

**Figure 3 PQQ-dependent methanol dehydrogenase (XoxF and MxaF) protein clades and abundances in the soil zones.** PQQ-dependent methanol dehydrogenase (XoxF and MxaF) protein tree containing sequences from the literature and experimental sequences in the soil zones with their corresponding relative abundance of normalized spectral counts from the proteomics results in a heat map.

for protein uptake and degradation, such as extracellular serine-type endopeptidases (annotated as pyrolisin-like serine protease and encoded with a N-terminal signal peptide), amino acid transporters, and cytoplasmic amidases and formamidases highly represented in the proteome. The findings parallel results for marine Thermoplasmatales

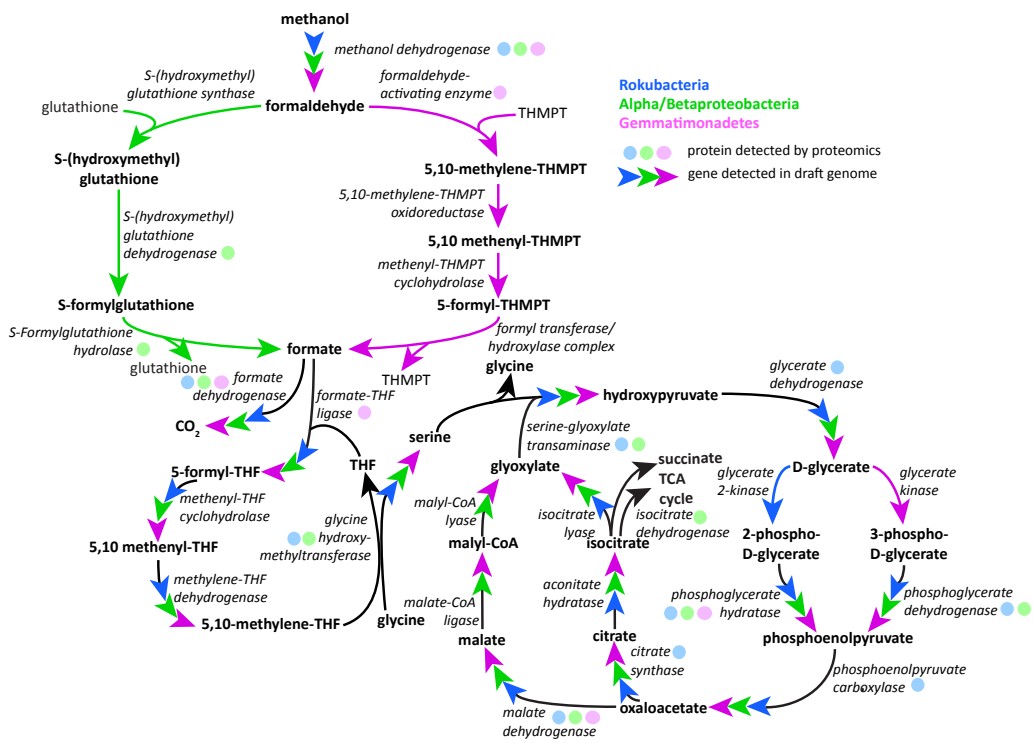

**Figure 4** **Putative active methylotrophy pathways in Rokubacteria, Proteobacteria, and Gemmatimonadetes.** The colored arrows indicate genetic evidence of proteins that catalyze the reactions that transform methanol into $CO_2$ or cell material and colored dots indicate representation in the proteome. Gemmatimonadetes (purple) and Proteobacteria can oxidize formaldehyde via the tetrahydromethanopterin and glutathione-linked pathways, respectively, while Rokubacteria (blue) cannot. Formate is incorporated by the tetrahydrofolate-linked pathway then converted to serine by glycine hydroxymethyltransferase. Serine and glyoxylate are used by serine-glyoxylate transaminase to produce glycine and hydroxypyruvate, which is then converted to D-glycerate. D-glycerate can be phosphorylated by two different kinases, glycerate 2-kinase in Rokubacteria and glycerate kinase in Gemmatimonadetes. Both phosphoglycerates are converted by different enzymes to phosphoenolpyruvate (phosphoglycerate hydratase or phosphoglycerate dehydrogenase) and then to oxaloacetate by phosphoenolpyruvate carboxylase. Gemmatimonadetes and Proteobacteria (green) contain but Rokubacteria lack the canonical malate-intermediated serine formaldehyde assimilation pathway but do encode citric acid/glyoxylate cycle genes that could assimilate carbon (citrate synthase) and regenerate glyoxylate (isocitrate lyase).

(*Lloyd et al., 2013*) and members of the TACK superfamily from groundwater (*Castelle et al., 2015*), and suggest a role for novel soil Archaea in protein degradation.

We investigated the distribution of metabolites in triplicate extracted soil water samples using a robust Liquid Chromatography-based Mass Spectroscopy (LC-MS) metabolomics workflow based on a previous study on samples from the meadow soils (*Swenson et al., 2015*). A total of 125 unique compounds were detected and quantified. These include sugars (one to six sugar residues per chain), sugar alcohols, amino acids, nucleotides/nucleosides, quaternary amines, osmolytes and several suspected sugar metabolites and derivatives (Fig. 5). Notably, most concentrations fall to zero at the base of the soil zone, an observation that suggests efficient scavenging of these compounds by microbial soil communities and not the sorption to mineral surfaces (*Fischer, Ingwersen & Kuzyakov, 2010*).

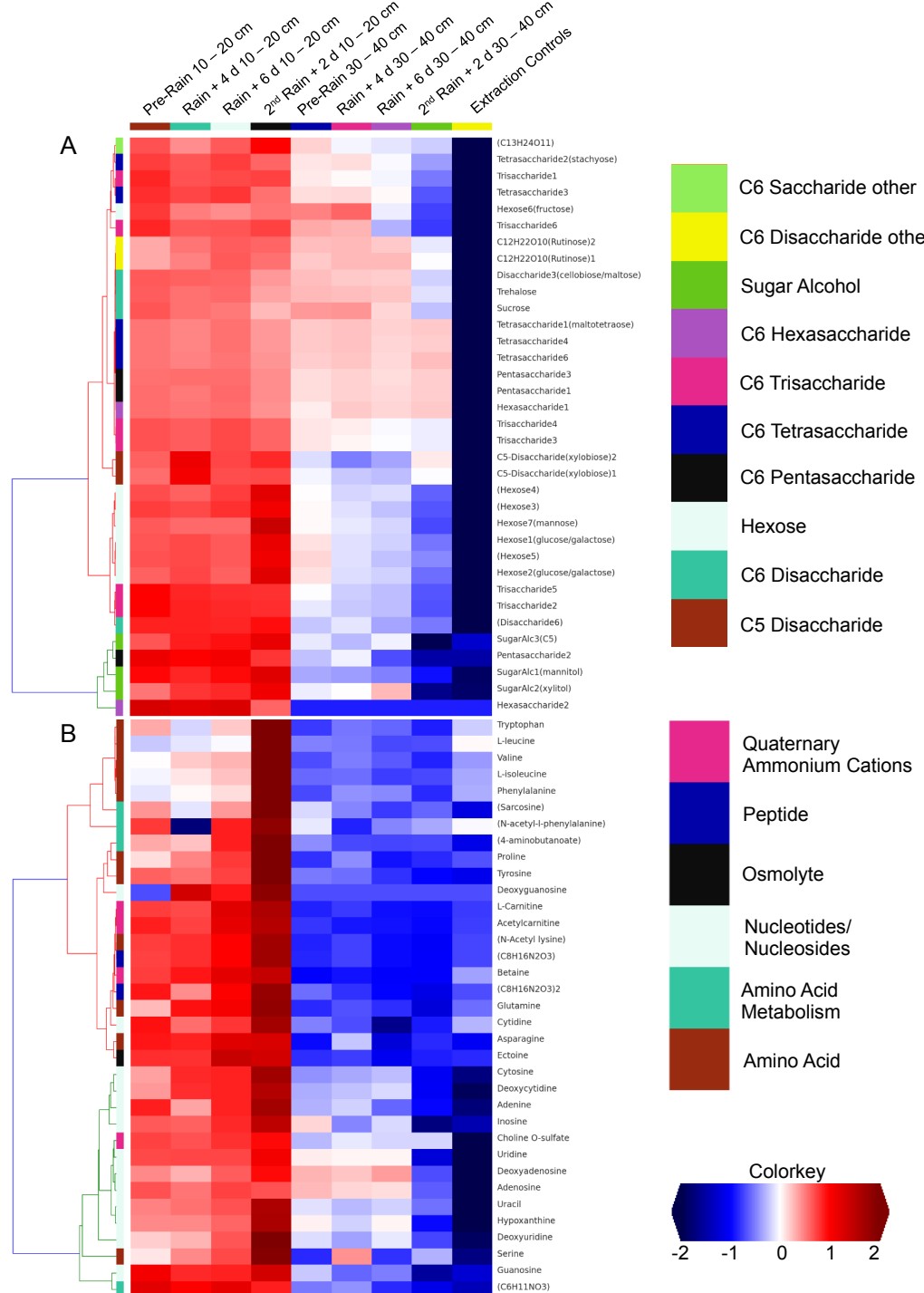

**Figure 5  Comparison of detected sugars and common nitrogenous compounds in Angelo soils at 10–20 cm and 30–40 cm.** (A) lists sugars and (B) lists nitrogen-containing compounds and the heat map indicates relative mean sample concentrations. Columns are ordered left to right by date. Row colors (left sides of plot) are based on chosen metabolite groupings. A clear decrease in metabolite abundances is observed with increasing soil depth.

Dissolved organic nitrogen, which is comprised of nitrogen-bearing molecules such as amino acids, represents an important nutrient source for soil microbial communities (*Jones et al., 2004*). The metabolomic data identifies dramatic increases in amino acid and nucleotide concentrations at 10–20 cm right after the second rain event (Fig. 5). The timing of this increase, immediately following a substantial rainfall event, points to the vertical transport of these compounds from the top 10 cm. However, the source and localization of their production were not determined in the current experiment. Notably, there was no observed accumulation of these compounds in the deeper soil (30–40 cm). This may be attributable to microbial activity at these depths. For example, the Thermoplasmatales and Bathyarchaeota archaea, which were highly represented in the 30–40 cm depth soil especially after rain events, have proteins for uptake and aerobic breakdown of peptides. Abundances of these proteins, which include dipeptide, oligopeptide, hydrophobic peptide, polar peptide transporters are high in samples collected after the first rainfall (Table S3A).

Interestingly, ammonia-producing formamidase is the most abundant protein identified for several archaea. Ammonia liberated by archaeal peptide degradation likely supports growth of SAGMCG and SCG Thaumarchaeota, which both have ammonia monooxygenases genes. A copper-containing nitrite reductase (NirK), along with cytochromes, sulfurtransferases and Fe-S proteins, were abundant in the proteome but NO reductases were not identified by proteomics (although they are encoded in the genomes). Overall the results suggest roles for Thaumarchaeota in both nitrification and denitrification

Breakdown of plant-derived organics can release sulfur compounds. For example, glucosinolates are sulfur-bearing organics that are produced by Brassicales, a widely distributed group of plants in the mustard family that were identified in this study's meadow. Degradation products of glucosinolates include sulfur-containing thiocyanate and isothionates. The Rokubacteria have genes of the Sox sulfur oxidation pathway, as do the first-described Rokubacteria described from groundwater (*Hug et al., 2016*), and some Sox proteins were identified in the proteome. Thus, these novel bacteria are inferred to play an important role in sulfur biogeochemistry in the sub-root zone soil during the rainfall-induced period of organic matter turnover.

Choline sulfate is an interesting sulfur-containing osmolyte identified in the metabolome. This compound is produced by many organisms, including plants and fungi, and is degraded to produce betaine via sulfatase enzymes. Betaine was also identified by metabolomics. Notably, sulfatases were observed in the proteomes of Actinobacteria, Chloroflexi, Alphaproteobacteria and Betaproteobacteria. Sulfate released from choline sulfate degradation may play an important role in soil bacterial sulfur metabolism (*Markham et al., 1993*). Choline sulfate and betaine both contain three methyl groups per molecule that are released upon the degradation to glycine and could contribute to the growth of the methylotrophic bacteria.

## DISCUSSION

Our study design aimed to deeply analyze soil microbial community composition and to detect genes and proteins from many organisms, including those at relatively low

abundance. This generated a soil metagenomic dataset of unprecedented size (>200 Gb) and complexity (>1,400 species). Given the massive data sizes involved in this research, we limited analysis to ten samples so we do not attempt to describe overall shifts in the patterns of microbial distribution during this time period. However, the spectrum of conditions provided access to a wider variety of genomes than would be provided if analyses target a single sampling location or time point. Also, we extracted DNA from large (~200 g) homogenized samples; this likely reduced the impact of spatial microheterogeneity and probably explains why overlap in community composition could be detected.

Only recently has genome reconstruction from soil (seven from permafrost by *Hultman et al. (2015)*, seventeen from enrichments by *Delmont et al. (2015)*, and 129 from prairie soil by *White et al. (2016)*) been achieved. In our study, no single organism represented >3% of the community. The extensive strain and within population variation likely explains the high level of fragmentation of genomes for some organisms (Tables S1 and S2). Even with these challenges, we reconstructed genomes from all the major lineages represented in the microbial communities. We attribute this result to the assembly of reads into large scaffolds, many of which could be binned because overlap in community composition over the sample series enabled binning using abundance pattern information. Scaffold assembly provided high quality ribosomal protein S3 sequences that were used to distinguish organisms at the species level, a phylogenetic resolution exceeding that which could be obtained by rRNA sequencing methods (*Sharon et al., 2015*). The predicted protein dataset provided the foundation for multi-omic analyses that yielded functional insights.

Most prior research on carbon cycling in soil has focused on microbial degradation of complex soil organic macromolecules, likely derived in part from plant biomass. Our metabolomics analysis suggest that the organic and nitrogenous substrates needed to sustain microbial life disappear from the soils relatively rapidly as few metabolites accumulate to measurable amounts in the deeper soil profile. Nitrogenous compounds, mostly free amino acids, were identified in soil after the second rain in the 10–20 cm zone yet were practically undetectable in the 30–40 cm zone. It is clear that the substrate availability between one 10 cm zone to the next is very different to that in the upper soil horizon, and will support different microbes. For example, the Verrucomicrobia and Actinobacteria are only abundant in the 10–20 cm depth interval and much rarer in the 30–40 cm depth interval. We also found abundant, diverse proteins involved amino acid and carbohydrate degradation and import in every partial to near-complete draft genome. Because we employed an untargeted proteomics, we could identify many thousands of proteins using a peptide database composed of full-length genes predicted from the samples' metagenomes.

Yet, interestingly the most abundant protein in the proteomics data was the PQQ-dependent methanol dehydrogenase from Gemmatimonadetes and Rokubacteria. In addition to complex carbohydrates, plant biomass and root exudates also provide an abundant source of methanol (*Sutton & Sposito, 2005*). Previously, only members of Proteobacteria, NC10, and Verrucomicrobia have been shown to be methylotrophs (*Chistoserdova, 2011*; *Op den Camp et al., 2009*). Methylotrophy was tentatively linked to Gemmatimonadetes only once before, when $^{13}C$ methanol containing compounds were

fed to a lake sediment sample and labeled Gemmatimonadetes 16S rRNA was identified (*Nercessian et al., 2005*). Methylotrophy has been described in aerobic lake sediments (*Costello & Lidstrom, 1999*), the phyllosphere (*Corpe & Rheem, 1989*; *Delmotte et al., 2009*) marine (*Radajewski et al., 2002*; *Stacheter et al., 2013*), and soil (*Eyice et al., 2015*; *Kolb, 2009*; *Radajewski et al., 2002*; *Stacheter et al., 2013*) environments. These studies found that methanol-oxidizing enzymes of Proteobacteria have micro- and nanomolar affinity for methanol, the highest activity occurring in the root-associated soil, and that methylotrophic communities thrive under the full range of plant diversity and soil pH (*Radajewski et al., 2002*; *Stacheter et al., 2013*). Further, methylotrophic methanogenesis can occur under aerobic conditions (*Hofmann et al., 2016*; *Karl et al., 2008*; *Metcalf et al., 2012*). However, in our study, no methyl-coenzyme M reductase complex (*mcrA*) gene was predicted in any dataset. Thus, methylotrophy is neither occurring in nor linked to co-occurring methanogens.

Notably, we observe a significant fraction of the microbial community (87 distinct organisms via rpS3 genes) belong to the as yet uncultured yet widespread phylum Bathyarchaeota (formerly known as the Miscellaneous Crenarchaeotal Group) (*Gagen et al., 2013*; *Kubo et al., 2012*). Bathyarchaeota have been identified by 16S rRNA studies of sulfate-methane transition zones and hypothesized as being involved in dissimilatory anaerobic methane oxidation coupled to organic carbon assimilation (*Biddle et al., 2006*). In a recent report, Bathyarchaeota were identified in metagenomic analyses of coal-bed methane well water. The genomes encoded a complete methanogenic pathway including an ancient *mcrA* (*Evans et al., 2015*). The four draft (71–91%) Bathyarchaeota genomes do not contain the genes required for methanotrophy or methanogenesis but encode oligopeptide import and amino acid degradation pathways, which were also abundant in the proteomics analysis. The large transporter diversity suggests substantial substrate flexibility in Bathyarchaeota (and also in Thermoplasmatales archaea). Thus, along with other community members including Thermoplasmatales, Bathyarchaeota likely contribute to degradation of nitrogen-containing compounds in the deeper soil. The findings underline the importance of genomic resolution, because metabolic roles of the soil Bathyarchaeota predicted based on phylogenetic information and previously published genomes would have been incorrect.

## CONCLUSION

This genome-resolved multi-omic study revealed many populations of little known bacteria and archaea in sub-root zone soil microbial communities. Our proteogenomic analysis yielded strong evidence for methanol oxidation in novel members of the Gemmatimonadetes and Rokubacteria phyla. These capacities have not been previously linked to organisms of these phyla, although Gemmatimonadetes are common members of soil microbial communities. Rokubacteria, on the other hand, have not previously been reported from soil, so the findings of this study contribute new information regarding microbial community composition as well as function. Removal of methanol and other small organic molecules from solutions draining from upper soil horizons by these

bacteria limits their availability for metabolism by organisms at greater distance from leaf-litter associated carbon sources. Although methanogeneis is not prominent in the studied grassland, such activities in other soils could restrict the supply of methanol to methylotrophic methanogens in deeper subsurface regions. We found that different microbes and metabolites are abundant in samples collected just 10 cm apart. Likely, the organisms are stratified by substrate availability, a pattern that results in part from the activities of organisms in the overlying soil regions.

## ACKNOWLEDGEMENTS

We would like to thank the rest of the members (and former members) of the Banfield lab for their help with and the development of various tools and reference libraries, and Dr. David Burstein for help with collecting soil samples. The sequencing was conducted by the US Department of Energy Joint Genome Institute, a DOE Office of Science User Facility, and Lawrence Berkeley National Laboratory.

### Funding

This work is supported by the Office of Science, Office of Biological and Environmental Research, of the US Department of Energy Grant DOE-SC10010566. The sequencing was conducted by the US Department of Energy Joint Genome Institute, a DOE Office of Science User Facility, and Lawrence Berkeley National Laboratory under Contract No. DE-AC02-05CH11231. The funders had no role in study design, data collection and analysis, decision to publish, or preparation of the manuscript.

### Grant Disclosures

The following grant information was disclosed by the authors:
Office of Science, Office of Biological and Environmental Research, of the US Department of Energy: DOE-SC10010566.
US Department of Energy Joint Genome Institute.
DOE Office of Science User Facility. Lawrence Berkeley National Laboratory: DE-AC02-05CH11231.

### Competing Interests

The authors declare there are no competing interests.

### Author Contributions

- Cristina N. Butterfield, Zhou Li and Peter F. Andeer conceived and designed the experiments, performed the experiments, analyzed the data, contributed reagents/materials/analysis tools, wrote the paper, prepared figures and/or tables, reviewed drafts of the paper.
- Susan Spaulding performed the experiments, contributed reagents/materials/analysis tools, prepared figures and/or tables.

- Brian C. Thomas performed the experiments, contributed reagents/materials/analysis tools, wrote the paper, prepared figures and/or tables, reviewed drafts of the paper.
- Andrea Singh performed the experiments, contributed reagents/materials/analysis tools.
- Robert L. Hettich, Trent Northen and Chongle Pan conceived and designed the experiments, contributed reagents/materials/analysis tools, wrote the paper, reviewed drafts of the paper.
- Kenwyn B. Suttle performed the experiments, contributed reagents/materials/analysis tools, wrote the paper, reviewed drafts of the paper.
- Alexander J. Probst performed the experiments, contributed reagents/materials/analysis tools, wrote the paper, prepared figures and/or tables.
- Susannah G. Tringe contributed reagents/materials/analysis tools, wrote the paper, reviewed drafts of the paper.
- Jillian F. Banfield conceived and designed the experiments, analyzed the data, contributed reagents/materials/analysis tools, wrote the paper, prepared figures and/or tables, reviewed drafts of the paper.

## Field Study Permissions

The following information was supplied relating to field study approvals (i.e., approving body and any reference numbers):

Angelo Coast Range Reserve Permission APP#27790.

## DNA Deposition

The following information was supplied regarding the deposition of DNA sequences:

Sequencing reads: ''Meadow soil samples from Angelo, CA genome sequencing and assembly'': SRA302421; Soil metagenome, BioProject PRJNA297196; and all data is also available in ggKbase: http://ggkbase.berkeley.edu/angelo_ncbi_2016/organisms.

## Data Availability

The raw data has been supplied as a Supplemental File.

## Supplemental Information

Supplemental information for this article can be found online at http://dx.doi.org/10.7717/peerj.2687#supplemental-information.

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
