# Peer review of "Proteogenomic analyses indicate bacterial methylotrophy and archaeal heterotrophy are prevalent below the grass root zone"

_PeerJ, doi:10.7717/peerj.2687_

## Round 0.1 · original submission · Major Revisions

Dear Dr. Butterfield,

I thank you and your colleagues for your submission and for your patience.

Along with my suggestions, I am happy to return to you two independent reviews of your study done by experts in the field who were kind enough to lend their time to evaluate it.

Overall, I find the study very exciting. It not only demonstrates that the state-of-the-art genome-resolved approaches progressively extends our reach to de novo reconstructed genomes even from most complex environments, but also shows the promise of investigating microbially mediated biochemical processes in the context of these novel genomes by combining multiple 'omics approaches.

In addition to reviewer comments, I would like to mention a number of points:

Reading the methods section, I wasn't completely sure whether the samples were co-assembled, or assembled individually and then reads were mapped to recover coverages of each contig in each sample. The reviewers have other questions about the methods used, which suggests to me that the methods section can be further improved to increase its clarity and coverage.

I also share the concern raised by both reviewers regarding the data availability. Raw sequencing data, as well as metaproteome, and metabolome data should be available to the community (including the reviewers). Also the URL provided in the manuscript, http://ggkbase.berkeley.edu/project_groups/angelo, is not accessible due to permissions.

I think the abstract could be better structured for a smoother reading experience. For instance, the sentence "The genomes represent all major phylotypes." seems to be cutting in between information about samples and the sampling strategy. The abstract is also missing the opportunity to highlight the power of this study. A sentence similar to the last sentence of the Introduction may have been useful to convey its strength.

I finally have a minor concern related to terminology: there are multiple instances where the text refers to genomes reconstructed from metagenomes as 'draft genomes'. The term draft genome is often used to describe incomplete assemblies of isolates, and makes it harder to appreciate the unique nature of most metagenome-assembled genomes. I think referring to these genomes as 'metagenome-assembled genomes', or even 'population genomes' (even though what a population means is also questionable in the context of microbiology) would set the stage better, and clarify what we are working with further. Although I am not necessarily asking for a change in the text, I wanted to bring this to your attention in case you would like to reconsider.

--

I would like to thank again to both reviewers for their time, and the effort they put into the evaluation of this work.

Reviewer 1 ·

Basic reporting

The authors have written a grammatically sound manuscript. No major comments/revisions are noted in this regard. In general, the introduction and background was sufficiently detailed to provide proper context. One instance where I felt a shifting of emphasis would be beneficial was in the description of metagenomics advancements in soil systems (paragraph 3) and the subsequent paragraph introducing the scope of work for this manuscript. I felt that the description of metagenomics advancements was too extensive considering that more could be said to emphasize the rationale for this work. For example, a more detailed presentation of information related to microbiological processes and dynamics in this specific habitat that are related to larger processes, such as carbon cycling, would be useful. Additionally, the intricate controls and interactions of soil minerals, precipitation events, and microbial activity remain relatively unaddressed in the introduction, as it stands. While the technical achievements of this work are laudable on their own, the biological insights gained into the microbiology of this system by leveraging these multiple omics datasets is arguably more important.

I often found the figures did not serve the text as well as I would hope. The first figure provided general community structure information, but it does not complement the text very well. As a reader I am drawn to the coloring of the bars, which indicate shared taxa across the samples, but I didn't find that the relevance of this information was a primary finding of this work based on the text. However, based on my reading of the text, I was interesting to understand how each sample varied in community composition. Something akin to a Beta-diversity plot might be more useful here. As for Figure 2, the results presented are hardly discussed in the text, and the figure does not really impress upon the reader any connection between ecological and evolutionary processes occurring across these samples. I’m not sure if, as a reader, I am supposed to note an interesting feature of the Gemmatimonadetes, or if this taxa was chosen simply because of their abundance. The remaining figures are much better at describing ecological and metabolic patterns within the data. The figure 5 color gradient could be changed to be more color-blind friendly.

I may have missed it, but I do not see a description of actions taken to release the raw data generated by this study (e.g., metagenome sequence data, metaproteome, and metabolome data).

Experimental design

The submitted research is original to the best of my knowledge, and the research question is clearly articulated. I have already commented on how the introduction section could better state the knowledge gap being addressed and the importance of the biological question being posed.

The experimental design seems logical. Biological replication of metagenomic and metaproteomic samples is desired, but this is still lacking in many studies. So, it would appear that the field as a whole accepts this compromise on behalf of costs.

A bit more description for some of the methods is needed in some places, in my opinion. Three areas stand out for me. First, a better description of the sampling design is needed in the methods. It was only after digging into the supplementals that I started to understand which samples were taken on which days. Confusion arises when the authors mention the use of only 10 samples in this study, which muddies an understanding of which samples were analyzed. Also in the results (line 274), 18 samples are mentioned. I’m assuming this is a typo, but it contributed to the confusion of which samples were taken, where replicates were collected from, and when each sample was taken (e.g., information on how many days before the first rain and when the second rain occurred is not described except in Figure S1). The second place that more information is needed is in the description of the metagenome assembly. It is not very clear unless the supplemental information is scrutinized whether the metagenomes were each assembled separately, or co-assembled with reads from individual libraries mapped back. Since it appears the former approach was taken, it remains unclear whether bins representing the essentially same organism (at the level of detectable phylogenetic resolution) are merged across samples, or are more or less redundant. Additional minor issues that should be addressed would be to include more details regarding the binning procedure (e.g., were coverage values weighted relative to tetranucleotide frequencies for the ESOM? how were contigs originally assigned a taxonomy prior to binning? how is bin contamination assessed?). The last section that should receive minor attention is regarding the metabolomics work. For example, is there a citation for the Northen Lab standards library that could be provided? Also, a more in-depth description of how the internal control was used and how the measured values of this control are interpreted in Figure 5 would be useful.

Validity of the findings

Two sample statistical tests could be used to more strongly support the results and conclusions of this study. The major conclusions seem valid and supported by the data. Some minor remarks regarding some discussion items that are not as robustly supported are noted below in the General Comments section.

Additional comments

Lines 24 - 25: Sentence is vague enough that it carries little weight. Proteogenomics itself still has problems with ambiguous links between true function carried out in soils and matching functions to the organisms. Its better than other techniques, but still lacking. Also, we do know a lot about soil respiration at the community level. So maybe adding some specifics to your statement would help to elevate it beyond vague generalities. Why is it important to study this process beyond the community level? Does answering ‘who’s doing what’ start to tell us something more about the system, which allows for a deeper understanding of process and mechanisms?

Lines 61 - 67: I’m not sure how evolutionary/adaptive pressures would explicitly provide a rationale for detection of novel organisms? Is it just because soil would select for high diversity? If so, you already stated that with your previous point.

Lines 109 - 112: These two sentences come without much context, almost as a preemptive defense of criticism, which would not be given at this point. You may want to consider placing more context prior to these statements, or moving them to the methods or discussion sections.

Line 244: Was no correction made for multiple hypotheses testing for all metabolites using ANOVA?

Lines 282 - 283: This statement is not based on an independent measure of abundance. How do you know if there are abundant organisms in your sample that do not have well-assembled genomes?

Lines 364 - 369: What specific finding is similar to the Lloyd paper? You mention amino acid transporters in the lead up, but this is in combination with other enzymes. The Lloyd reference suggests archaea are specifically involved in protein degradation, not cyclic organics or polysaccharides. Although, you could argue these are all complex carbon substrates, the mechanisms of degradation of each are very different, so the comparison is somewhat misleading.

Lines 370 - 378: A better description of how the physical and biological processes might govern some of the specific metabolite patterns would be useful. The metabolomics data is interesting, but difficult to interpret in an ecologically relevant way. For instance, it is not quite clear why amino acids would generally be less abundant in the deeper samples. Is this suggesting that these metabolites are released at both depths, but the deeper communities are more efficient at turnover? A more rigorous analysis of this data in light of what is known of the presumed biological activities and physical processes is needed.

Line 404 - 411: Does a correlation exist between where these metabolites are detected and the presence of enzymes used to metabolize them? I.e. is there an inverse correlation between choline sulfate and betaine, with sulfatase enzymes being positively correlated with betaine? Maybe this is too simplistic, but it is a starting point for understanding these metabolic patterns from a biological perspective?

Line 419: How is overlap being measured? The average sample count for 652 taxa is ~4, which would support the idea that your samples are segregated by community composition differences between soil depths? Without some replicates more data would be needed to assess this statement properly.

Line 424: What is the evidence for fragmented genomes? N50 of the supplemental table?

Line 428 - 430: It may be true that the genome recovery provides better phylogenetic resolution than rRNA sequencing, but what is the evidence presented here? No rRNA sequencing was performed, so a direct comparison between that method and the metagenomics analyses to assess phylogenetic resolution can not be performed. rRNA sequencing provides pretty good resolution, and, conversely, there are many times that metagenomic bins can not be confidently assigned at high resolution.

Line 433: Can you make conclusions about the speed of metabolite utilization without time course experiments. What is the rate of transport of the water across the soil?

End of discussion and conclusion: Methanogenesis is discussed. Is there evidence that these soils turn anaerobic?

Supplemental data: Are all of these tables and figures referenced and and numbered accordingly throughout the text? (E.g., I am not finding a reference for Figure S1, which seems to have relevance for developing an understanding of water transport across soil depths.)

Table S2ab: Better formatting would aid in reading this table.

Figure S2: What are the colored branches indicating? I am assuming taxonomic groupings, but it seems a bit inconsistent, with black branches interspersed here and there, and I’m not sure why. This could be explained in the legend better. Also, the under-representation of gamma-proteobacteria and bacteroidetes is very striking to me. Is this to be expected in these soils?

Reviewer 2 ·

Basic reporting

I did not see information about public deposition of the proteomic mass spec data or processed Sipros results; such data should be placed in a public repository such as proteomeXchange.

Experimental design

I would have liked to have more information about how the protein sequence database for Sipros searching was constructed -- its size, number of sequences, &c. Was all ~20Gb of sequence data (or at least the predicted protein-coding portions) included? Were only full-length predicted sequences used? Was any data other than the metagenomes from these samples used in the search database? The proteomics results are critically dependent on the search database, so more specifics on how this was done would be helpful. The database itself should also be made available, either as a supplement or via a repository.

Validity of the findings

No comments, meets guidelines.

---

## Round 0.2 · Minor Revisions

Dear authors,

Thank you for your revised manuscript which addresses most of the reviewer concerns raised during the initial round of review. I am sending the manuscript back to you for minor revisions this time, since there are some remaining points to address as highlighted by the reviewer #1.

I thank you for your attention, and the reviewer #1 for their effort to improve the study.


Best,

Reviewer 1 ·

Basic reporting

The main item of concern (link between text and figures) has been sufficiently addressed.

Experimental design

Previous comments I had regarding experimental design have generally been addressed. I still do not believe that a strong rationale has been provided for not performing an FDR correction on the metabolomics data. Having a relatively low number of comparisons does not justify not using corrective methods (hundreds of tests, although less than 1000s, is still not trivial with regard to assessing the validity of the results), since these methods explicitly account for sample size in the FDR calculation.

Validity of the findings

My comments have been sufficiently addressed.

Additional comments

Based on the data presented, I still find it difficult to believe the conclusions made regarding metabolite transport and utilization. The conclusions made by the authors are not unreasonable, but, in my opinion, they need to be more thoroughly supported by additional data, as alternative explanations for the observations can not be ruled out. E.g., it is suggested that relatively high abundance of metabolites that increase in 10-20 cm soils over subsequent sampling days vs. low abundance in deeper soils taken at the same time support transport of metabolites between the two depths. Could this observation not also be supported by a system where no transport of metabolites between depths is occurring? How could one tell the difference? Is there a metabolite that is known to be slowly degraded, which could be used as a "tracer" for water transport between depths? Actually, I am now realizing that Fig. S1 does show water moisture content increases over time at all depths. What are the significant differences here? Which "Days" relate to the time of sampling in this figure? How does this information impact your conclusion about transport of metabolites? This seems like a good starting point for developing an argument.

Also, the presence of carboxymethylenebutenolidase and a metal-dependent hydrolase in the proteome of archaeal populations is far from definitive evidence for protein degradation activity in the Archaea. I'm not sure why the first enzyme is mentioned in this context. In searching for information on this enzyme I found nothing suggesting its role in protein degradation. If I'm wrong I'm happy to reconsider my comment, since I do not have much prior knowledge of this enzyme's activity.

My overall suggestion to the authors regarding the metabolomics data would be to more rigorously consider the conclusions being made by better supporting them with additional outside references or with unambiguous data from this study. References showing the role of soil archaea in amino acid cycling, citations supporting the conclusions that these enzymes are specifically involved in protein degradation, and data from this experiment that supports water transport processes would make sense here.

---

## Round 0.3 · accepted · Accept

Thank you very much for the revised manuscript and for addressing those final points raised by the reviewer. I also thank the reviewers for their time and attention.